# Oriented Tapes of Incompatible Polymers Using a Novel Multiplication Co-Extrusion Process

**DOI:** 10.3390/polym14183872

**Published:** 2022-09-16

**Authors:** Xinting Wang, Erik J. Price, Gary E. Wnek, Andrew Olah, Eric Baer

**Affiliations:** Center for Layered Polymeric Systems (CLiPS), Department of Macromolecular Science and Engineering, Case Western Reserve University, Cleveland, OH 44106, USA

**Keywords:** multiplication co-extrusion, PP/HDPE blend, post-extrusion orientation, PP/HDPE interface, mechanical properties

## Abstract

Continuous tapes of polypropylene (PP) and high-density polyethylene (HDPE) were produced using a novel multiplication co-extrusion process. The structure of the PP/HDPE tapes consists of co-continuous PP and HDPE domains aligned in the extrusion direction, forming a fiber-like composite structure with individual domain thicknesses of 200–500 nm. This unique structure created a significantly large contact interface between the polymer domains. AFM images suggest strong interfacial interactions between incompatible PP and HDPE domains. Orientation at 130 °C was possible due to the enhanced adhesion arising from epitaxial crystallization and the large interfacial area. The modulus, tensile strength, and orientation factor of the oriented composite tapes increased as the draw ratio increased. The existence of two independent shish kabab-like morphologies in the oriented tapes at different draw ratios was indicated by the appearance of two melting peaks for each material. After one-step orientation at 130 °C to a draw ratio of 25, the moduli of the oriented tapes increased to approximately 10 GPa, and the tensile strength increased to approximately 540 MPa. These oriented tapes are stiffer and stronger than commercial tapes and do not fibrillate during the orientation process indicating some interfacial interaction between the domains.

## 1. Introduction

Numerous studies were conducted on incompatible polymer blends, with and without coupling agents. These studies illustrate the vast area of structure–property–processing relationships by investigating carefully designed macromolecular architectures. However, there are relatively few fundamental studies on the uniaxial orientation of these systems for producing stiff and strong tapes. Gallagher et al. [1] illustrated that blends of polyethylene (PE) and polypropylene (PP), when highly oriented, yielded very stiff tapes, even without the use of compatibilizers. Recently, Schmidt et al. [2], using conventional blend processing methods, studied the room temperature orientation of PE and PP blends that were subsequently annealed at temperatures slightly above the melting point of PE. The result of this annealing is that the PE chains were re-oriented perpendicular to the draw direction, suggesting epitaxial crystallization growth of the PE by the PP domains [3,4,5]. Conversely, Bartczak et al. [6] provided evidence that PE can nucleate PP by studying the primary nucleation of the spherulites. It was observed that the primary nucleation mechanism was strongly dependent on the crystallization temperature. These observations suggest that, in a multicomponent incompatible system, either PE or PP can serve as an epitaxial template.

These studies suggested that the interface between PE and PP can promote the adhesion required to create PP/HDPE tape materials with superior properties. Additionally, sufficient melt mixing is required to produce large contact surfaces between the two components. This idea was exploited by Polaskova et al. [7], who utilized a unique extrusion die combined with counter-rotating twin-screw extruders. A micro-fibrillar morphology was produced in the blend without further post-processing orientation. It was proposed that upon the post-extrusion orientation of these micro-fibrillar blends, tapes with enhanced mechanical properties could be produced.

The goal of this study was to produce PE/PP blend tapes using a previously reported continuous nano-fiber co-extrusion technique [8,9,10,11,12]. This technique was designed to systematically produce composite tapes with an enhanced interfacial area between the two immiscible polymers. This large surface area promoted contact between the two incompatible polymers, thereby promoting adhesion between the domains. PE and PP were chosen for this study because both polymers can be oriented into strong fibers and, as previously noted, are known to crystallize epitaxially under specific conditions.

## 2. Materials and Methods

### 2.1. Materials

Polypropylene (PP) (Exxon Mobil 2252E4) and high-density polyethylene (HDPE) (Dow Elite 5960G) were used in this study. PP is a homopolymer with a density of 0.900 g/cm^3^. HDPE was produced from metallocene catalysts with a density of 0.962 g/cm^3^.

### 2.2. Co-Extrusion of Tapes

PP/HDPE tapes with a 50/50 volume fraction were produced from a co-extrusion and multiplication system, shown in Figure 1a. The cross-sectional structure of the polymer melt after multiplication is illustrated in Figure 1b.

Equal amounts of PP and HDPE were extruded at 250 °C from two single-screw extruders and fed into a feed block, producing two horizontal layer configurations. In Step I, the PP/HDPE two-horizontal-layered melt is aligned vertically after passing through the first vertical multiplier. The melt continued through 17 additional vertical multipliers to achieve a calculated structure with 2^18^ or 262,144 vertical layers, as shown in Step II. Next, the melt proceeds through five horizontal layer multipliers, as shown in Step III. The melt flow, when passing through the horizontal multiplier, was cut vertically in the center of the flow, producing two independent melt flows. These two independent melt flows were then combined on top of one another. After passing through one horizontal multiplier and assuming that the melt was cut ideally in the center, the vertical number of layers decreased by half and the horizontal layer number increased by two. The aim of the five horizontal multipliers is to obtain a smaller domain size in the thickness direction. After passing through five horizontal multipliers, a calculated structure with 8192 horizontal and 32 vertical domains was produced, as shown in Figure 1c. By calculation, there were 8192 alternating PP and HDPE domains in one horizontal layer and 32 layers in the vertical direction. Finally, the melt was processed into a thin tape using a 3 in. wide exit die, as shown in Step IV. The tape was collected on a chill roll at 90 °C. The tape thickness was controlled by the speed of the chill roll to approximately 200 µm. Based on a 60 mm tape width, the theoretical individual domain size is 7.3 µm wide and 6.2 µm thick for both the PP and HDPE domains.

PP and HDPE control tapes were produced using the same process and conditions as the PP/HDPE (50/50) tapes. Both the A and B extruders were used to produce PP or HDPE control tapes. The tape dimensions for the PP and HDPE control tapes were the same as the PP/HDPE (50/50) blend tapes.

### 2.3. High-Temperature Orientation of Tapes

PP/HDPE (50/50) tapes, PP control tapes, and HDPE control tapes were cut into 3 cm × 8.2 cm rectangular shapes before drawing. The tapes were placed in an environmentally controlled chamber and drawn on a mechanical testing machine (MTS Alliance RT/30) with a gauge length of 12 mm and at a strain rate of 1000%/min. An 80-grit emery cloth was attached to the jaw faces of MTS grips to avoid slipping during drawing. The temperature of the chamber was set to either 130 °C or 110 °C. The tape was heated for 10 min before drawing, to equilibrate the temperature within the tape.

### 2.4. Tensile Measurements

The oriented PP/HDPE tapes and PP control tapes were embedded in Emery cloth using Super Glue 15187 before the tensile measurements. The oriented HDPE tapes were embedded in the Emery cloth using a J-B Weld 50101 MinuteWeld™ Instant-Setting Epoxy before the tensile measurements. The aim of this sample preparation step was to avoid slipping inside the grips during measurement. The mechanical properties of the oriented tapes were tested at room temperature at a draw rate of 100%/min by using Intron 5965 series universal testing system in accordance with ASTM D882.

### 2.5. Structural Characterization

Differential scanning calorimetry (DSC) measurements were performed using sealed aluminum pans on a TA Instruments Q2000 DSC unit. The DSC samples were heated from 25 °C to 200 °C and then cooled to 25 °C at a cooling rate of 10 °C/min. A two-minute isothermal procedure was conducted before heating or cooling.

The crystal orientations of the PP and HDPE domains were characterized using a Rigaku 2D wide-angle X-ray diffraction (WAXD) unit with a Cu Kα radiation source (wavelength λ = 0.1542 nm). The sample-to-detector distance was 125 mm. The beam exposure time for each sample was 2 h.

Morphological characterization was conducted using atomic force microscopy (AFM). The tape was embedded in Loctite Heavy Duty Epoxy and cured for 24 h at room temperature. The embedded tape was cut along the extrusion direction at −120 °C using a glass knife in a microtome (Leica EM UC7 Ultramicrotome) unit with a cryo-chamber. AFM images were collected using a scanning probe microscope (Digital Instruments Nanoscope IIIa) with a rectangular tapping mode probe. The tip radius of the probe was 10 nm. The spring constant was 50 N/m and the resonance frequency was 320 kHz.

## 3. Results

### 3.1. Characterization and Properties of Unoriented Tapes

Previously, a novel solid-state structure of the 50/50 PE/PP system was described using a unique surface-generating co-extrusion methodology, as shown in Figure 1. This co-extrusion design was used to create two-component battery separators [11] and other novel micro/nano-fiber systems [13]. These polymer systems were processed by taking advantage of the low adhesive properties of incompatible polymers, such as Surlyn/polystyrene, polyamide 6/polyethylene terephthalate, and polyethylene/polypropylene. In all these cases, delamination was achieved in the micro/nanofibrillar structures by either solvent extraction of one of the phases or by the impingement of a high-pressure water jet.

The co-extrusion process in this work produced uniquely structured PP and HDPE tape, which comprised a cross-sectional matrix of 8192 × 32 or 262,144 individual PP and HDPE domains, respectively. The co-extrusion multilayer fiber process produces fiber-like semi-continuous domains. Three tapes were used in this study. In addition to the 50/50 PP/HDPE blended tapes comprising approximately 263,000 distinct PP and HDPE domains aligned in the extrusion direction, two control tapes comprising PP and HDPE, respectively, were produced using the same multilayer co-extrusion process.

#### 3.1.1. Morphology

The AFM images shown in Figure 2a–c show the as-extruded tapes containing a 50/50 blend of PP/HDPE. These images show elongated domains oriented along the extrusion direction. The lighter image is the PP and the darker image is the HDPE. The individual domain widths varied considerably between 200 and 500 nm. The red dashed line in Figure 1c indicates the approximate boundary between the PE and PP domains, indicating a large interfacial surface contact area. This suggests the possibility of strong interactions between the crystal phases of the PE and PP domains. Jorden et al. [14] previously proposed similar interactions as “entangled crystals”, depending on the polymer grade used in their study. Furthermore, epitaxy-generated interactions between PE and PP crystalline phases are also conceivable [3,4,5] since both polymers can crystallize at a similar temperature.

Further examination of the AFM images in Figure 2c indicates that, within the individual PP/HDPE domains, numerous alternating lamellar structures can be observed perpendicular to the extrusion direction. This indicates the presence of shish kabab-like structures possibly in both the PP and HDPE domains.

#### 3.1.2. DSC and WAXD Characterization

The DSC thermographs and WAXD patterns of the PP, HDPE, and 50/50 PP/HDPE tapes are shown in Figure 3 and Figure 4, respectively. These data illustrate that the individual thermal and crystallographic properties of the PP and HDPE components were retained in the co-extruded tape, indicating the distinct incompatibility of the two polymer phases.

The crystallinity of the PP control tape and HDPE control tape was 39% and 63%, respectively. In the 50/50 PP/HDPE tape, the crystallinity of PP was 35% and the crystallinity of HDPE was 54%. Compared to the controls, the crystallinity of both PP and HDPE decreased slightly in the 50/50 PP/HDPE tape. As shown in Figure 3b, the crystallization peak of the PP control tape was 117 °C and that of the HDPE control tape was 122 °C. Only one crystallization peak at 120 °C was observed in the 50/50 PP/HDPE tape. This indicates that both PP and HDPE crystallize at a similar temperature, which could explain why the 50/50 PP/HDPE tape has slightly lower crystallinity compared with the controls.

As shown in Figure 4, the WAXS patterns of all undrawn tapes have isotropic ring reflections. The PP control tape has six reflections (Figure 4a) for the monoclinic structure: (110), (040), and (130) along the hk0 layer line, (022) along the hk2 layer line, and two coincidental reflections (041) and (111). Two reflections were observed for the orthorhombic structure of HDPE (Figure 4b) corresponding to the (110) and (200) reflections. The combined reflections of PP and HDPE did not change in the 50/50 PP/HDPE tape (Figure 4c), indicating that the individual crystal structures did not change in the blend tape and that the tape was composed of two distinct phases.

#### 3.1.3. Mechanical Properties of the Unoriented Tape

The room-temperature mechanical properties of the as-extruded PP and HDPE control tapes and 50/50 PP/HDPE tape were determined in accordance with ASTM D882. The draw rate was 100%/min, and all tests were conducted in triplicate. Figure 5 illustrates the representative stress–strain curves for the three different samples. The average values of the modulus and yield strength for the HDPE control were 0.56 ± 0.02 GPa and 14.4 ± 0.4 MPa, respectively. The modulus and yield strength averages for PP control were 0.75 ± 0.13 GP and 19.7 ± 0.5 MPa, respectively. The 50/50 PP/HDPE blend had averages for the modulus and yield strength of 0.75 ± 0.07 GPa and 19.3 ± 2.0 MPa, respectively.

These data show that the mechanical properties of the 50/50 PP/HDPE co-extruded tape were similar to those of the PP control tape and higher than those of the HDPE tape. This re-affirms, as shown in the AFM image, that the interfacial region between the PP and HDPE domains exhibits good adhesion between the PP and PE domains.

These results illustrate that a unique multicomponent morphology can be produced by utilizing a novel multiplication and co-extrusion process between two incompatible polymer components. This process allows the domain morphology of the tape to be controlled based on process design. Furthermore, this process allows the production of very small multicomponent domain morphologies with extremely high domain interfacial areas.

### 3.2. Orientation of Tapes

It is well known that the solid-state orientation of PE and PP into stiff and strong tapes depends on the precursor melt structure and cooling and strain rate conditions. The processing conditions required to obtain optimum mechanical properties are dependent upon the specific structural characteristics of the tape during the processing stage. This implies that a novel approach for creating a PP/PE system prior to orientation is necessary to achieve enhanced properties. The unoriented tape in this study revealed a previously discussed complex structure showing inter-penetration at the interface between the two components. This was a result of the novel extrusion process, which enabled the high-temperature orientation of the tape comprising two incompatible crystalline polymers.

#### 3.2.1. Mechanical Behavior during Orientation

Figure 6a,b illustrate the mechanical behavior during orientation of the 50/50 PP/HDPE tape at 130 °C. The data summarized in Table 1 follow the definitions in Figure 6 of modulus (i.e., 2% secant modulus), yield stress, work hardening slope, and draw ratio at fracture onset.

As shown in Table 1, owing to non-uniform deformation at 110 °C during the orientation process, HDPE cannot be successfully oriented since the work hardening region is very small and the fracture onset occurs at the relatively low draw ratio of 12. Similarly, PP shows the fracture onset at 110 °C and a low draw ratio of 12. The draw temperature (110 °C) was too far from the melting point of PP for efficient orientation in the solid state under uniaxial tension. This stress state induces dilation, which causes cavitation and subsequent premature failure at low draw ratios.

It is interesting to note that 50/50 PP/HDPE can be uniformly oriented at 130 °C to a draw ratio of at least 25 with a work hardening modulus of 0.3 MPa. This could only be achieved because of the strong adhesive behavior between the PP and HDPE domains. The HDPE control tape could not be successfully oriented at 130 °C since this temperature was too close to the melting point. The PP control tape could be drawn to a very high draw ratio of 29 without fracture at this temperature. Surprisingly, the PP/HDPE 50/50 tape could also be drawn to a high draw ratio of 25 with a work-hardening slope similar to that of the PP control tape. In the PP/HDPE 50/50 tape, polyethylene could be oriented near the PE melting point, where considerable work hardening occurs prior to the onset of fracture. This indicates that the PP domains contribute to the HDPE domain orientation arising from the strong interfacial adhesion within this complex co-continuous domain morphology.

#### 3.2.2. Mechanical Behavior of Oriented Tapes at Room Temperature

The 50/50 PP/HDPE tapes oriented at 130 °C were used for further investigation. HDPE was oriented near its melting point and PP at a desirable orientation temperature, where considerable work-hardening occurred prior to the onset of fracture. Six different draw ratios (DR) were selected to further investigate the structure–property relationships of tapes oriented at 130 °C: (1) DR = 1.25 at the yield point; (2) DR = 3.4 at the end of the necking region and beginning of the strain-hardening region; (3) DR = 8.5, 14 in the middle of the strain-hardening region; (4) DR = 20 at the end of the strain-hardening region; (5) DR = 25 at the maximum draw ratio before fracture.

Figure 7 illustrates typical stress–strain curves obtained at room temperature for tapes oriented at 130 °C to various draw ratios. The mechanical properties of the tapes are shown in Table 2. At a draw ratio of 25, tapes were produced with a modulus of approximately 10 GPa, a tensile strength of approximately 540 MPa, and an average strain at break of 10%. Comparisons to commercial tapes are shown in Table 3.

#### 3.2.3. Structural Characterization of Oriented Tapes

The DSC curves for the five different draw ratios of the PP/HDPE 50/50 tapes are shown in Figure 8a–e. Prior to the necking region, up to a draw ratio of 3.4, the DSC traces indicated that no significant morphological changes had occurred. However, at a draw ratio of 8.5, which is in the middle of the work-hardening region, two crystalline peaks were observed for HDPE: one at 130 °C and the other at 142 °C, and also, two crystalline peaks were observed for PP: one at 161 °C and the other at 168 °C. This suggests that both the PP and HDPE crystalline regions are each configured into shish kabab morphologies. The lower melting temperatures originate from the oriented lamella (kababs) and the higher melting temperatures originate from the extended core chains (shishes). These two crystalline morphologies interact at the interface as shown in Figure 2c. Bashir et al. [15] observed similar melting behavior for oriented polyethylene. This observation was also interpreted as oriented shish kabab morphology. Similar observations for PP were made by Mi et al. [16] using a novel tape-forming procedure in which oriented polypropylene was composed of shish kababs and quantitatively related this morphology to the mechanical behavior.

The DSC data became more complex in the work-hardening region at draw ratios of 14 and 20. In the PP/HDPE tape, the 142 °C peak for HDPE, attributed to higher chain extension, disappears entirely at a draw ratio of 20. This indicates that when HDPE is drawn at 130 °C in the 50/50 PP/HDPE tape, sufficient chain mobility exists to recrystallize into a lamellar configuration. The overall crystallinity of HDPE remained constant during orientation. However, as shown in Figure 9, the overall crystallinity of PP increased almost linearly as the draw ratio increased. In addition, the two melting peaks of PP increased and sharpened significantly. As the draw ratio increased, the high-temperature PP peak increased, indicating an increase in the concentration of extended chain PP fibrils (shish). The extended chain core fibrils act as nuclei for lamellar crystallization of the rest of the PP, thus increasing the overall crystallinity of PP.

The wide-angle X-ray results in Figure 10 show that both HDPE and PP continued to be oriented further with increasing draw ratios. Figure 11a shows Herman’s orientation function f_c_ based on the PP (110) and PP (040) planes. The first indication of enhanced crystal orientation is at a draw ratio of 3.4, which is the onset of the work-hardening region. The reflections of both PP and HDPE become sharper as the draw ratio increases. This surprising result indicates that HDPE is oriented owing to its interfacial adhesion to PP at 130 °C. This suggests that the HDPE domain interface adhered to the PP domain and followed PP orientation upon drawing as observed in Figure 2c. As expected, considerable orientation occurred in the work hardening region. The orientation function, f_c,_ for PP increases from 0.64 to 0.97 between the draw ratios from 3.4 to 25. This complements the DSC observations of the sharpened melting peaks.

Figure 11b shows that both the modulus and tensile strength increased linearly in the region between a draw ratio of 3.4 and 25. A comparison with Figure 11a shows that the mechanical behavior correlates linearly with the orientation function. Therefore, the effect of orientation on mechanical behavior can be broadly described in two regions. At low orientations (i.e., draw ratios from 1 to 3.4), the increase in the tape modulus and tensile strength is primarily due to the orientation of the hierarchical morphologies in both the HDPE and PP domains, as shown in Figure 2. The WAXD pattern shown in Figure 10a,b indicates that considerable orientation occurred between DR 1.25 and 3.4 in both polymers. However, at the draw ratio of 3.4, (Figure 8a,b), no changes in melting behaviors were observed in the DSC thermograph. In addition, the crystallinity of PP increased only slightly when the draw ratio increased from 1.25 to 3.4. This indicates that no extended chain crystals were generated, although the orientation of the PP and HDPE crystals occurred. As the draw ratio increased to 8.5, changes in the crystalline morphology previously described in DSC observations occurred for both polymers. The formation of the shish kabab structure of PP in the work-hardening region accounts for these important enhancements in both the modulus and tensile strength of the tapes.

## 4. Discussion

### Comparison to Other Materials

Relative to previous investigations that employed the same processing technique on LDPE/PP tape [8] and PA6/PEO tape [9], the 50/50 PP/HDPE tape in this study has a higher modulus and tensile strength. Comparisons are shown in Table 3. Common, widely used, commercial PP strapping materials have moduli and tensile strengths of about 2.2 GPa and 300 MPa, respectively. However, the 50/50 PP/HDPE tape in this study when oriented to a draw ratio of 25 at 130 °C had a modulus almost 5 times higher and 2 times higher tensile strength than that of the PP strapping material. Other PP composites found in the literature produced by other processes [17,18] did not show significant improvement in mechanical properties. Tapes produced by simple melt-mixing of 75 wt% HDPE and 25 wt% PP, upon drawing yielded a modulus of 3.5 GPa [17]. However, this tape was observed to fibrillate prematurely. An all-polypropylene composite [18] produced by the co-extrusion of PP with PP co-polymers and employing a complex orientation procedure had a modulus of 3.8 GPa at a draw ratio of 25.

The multilayer co-extruded composite 50/50 PP/HDPE tape described in this work, when drawn in a one-step orientation procedure at 130 °C did not fibrillate during orientation. In addition, the mechanical properties improved considerably compared to those of commercial PP tapes. Both of these observations can be attributable to the crystalline phase interfacial interactions between the incompatible PP and HDPE domains.

## 5. Conclusions

A novel multiplication co-extrusion process was used to produce tapes from two incompatible polymers. A unique feature of this process is the generation of an extremely large interfacial contact region between the two components during extrusion. Stiff and strong tapes were made by combining an equal amount of polyethylene (HDPE) and polypropylene (PP), which are known to be incompatible and normally lead to poor interfacial adhesion between each component.

Particular emphasis in this study was placed on identifying the optimum processing conditions. Tapes exhibiting strong mechanical properties were produced by orienting tapes at 130 °C. After one-step orientation at 130 °C to a draw ratio of 25, the moduli of the oriented tapes increased to approximately 10 GPa, and the tensile strength increased to approximately 540 MPa. These tapes were stiffer and stronger than various commercial tapes and did not fibrillate during the orientation. At this temperature, the soft polyethylene domains are oriented by constraining between the polypropylene domains, which also serve as a crystallization template [15]. In addition, at this temperature, polypropylene is oriented in the solid state, as evidenced by the considerable increase in crystallinity.

AFM analysis of the tape showed an irregular interface between the HDPE and PP domains, indicating a large interfacial surface area between the two components. The enhanced adhesion between the incompatible HDPE and PP domains could be a result of epitaxial crystallization between the two independent HDPE and PP shish kabab morphologies since both HDPE and PP were observed to crystallize at approximately the same temperature.

Upon orientation, the DSC results at various draw ratios showed that each polymer exhibited two melting temperatures. The high melting temperature was attributed to the extended chain crystal morphology and the low melting temperature to the lamellar crystalline morphology. This indicated the existence of a shish kabab-like morphology. Further studies with incompatible crystalline polymers are required to confirm these observations.

## Figures and Tables

**Figure 1 polymers-14-03872-f001:**
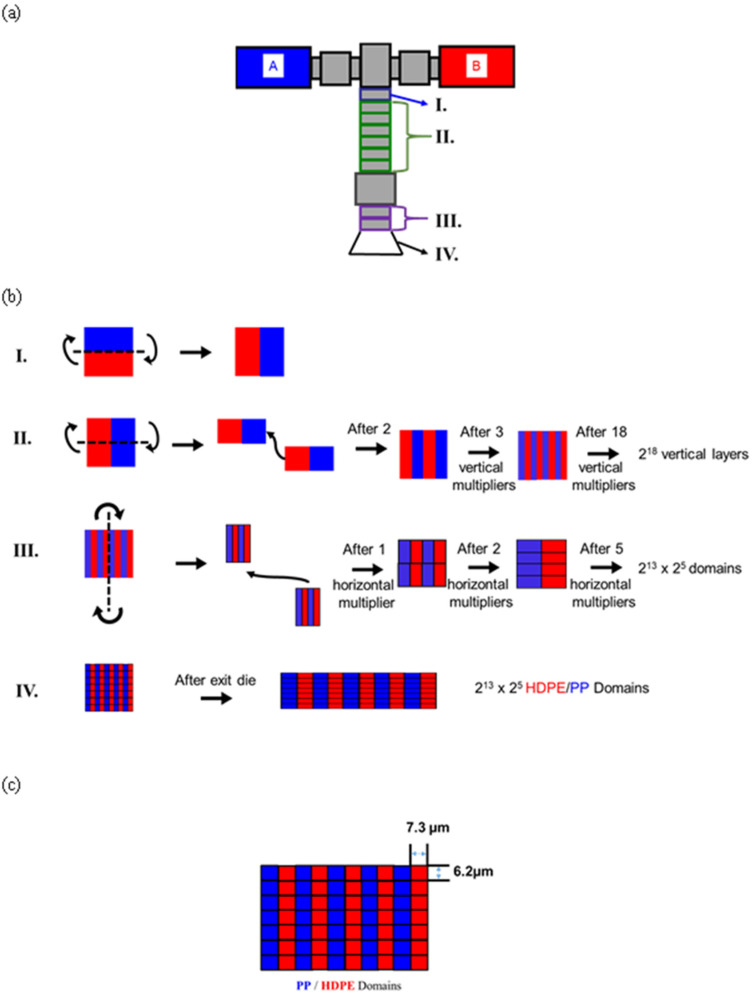
Schematic illustration of (**a**) co-extrusion and multiplication system; (**b**) theoretical cross-sectional melt structure during different multiplication steps; (**c**) theoretical tape structure.

**Figure 2 polymers-14-03872-f002:**
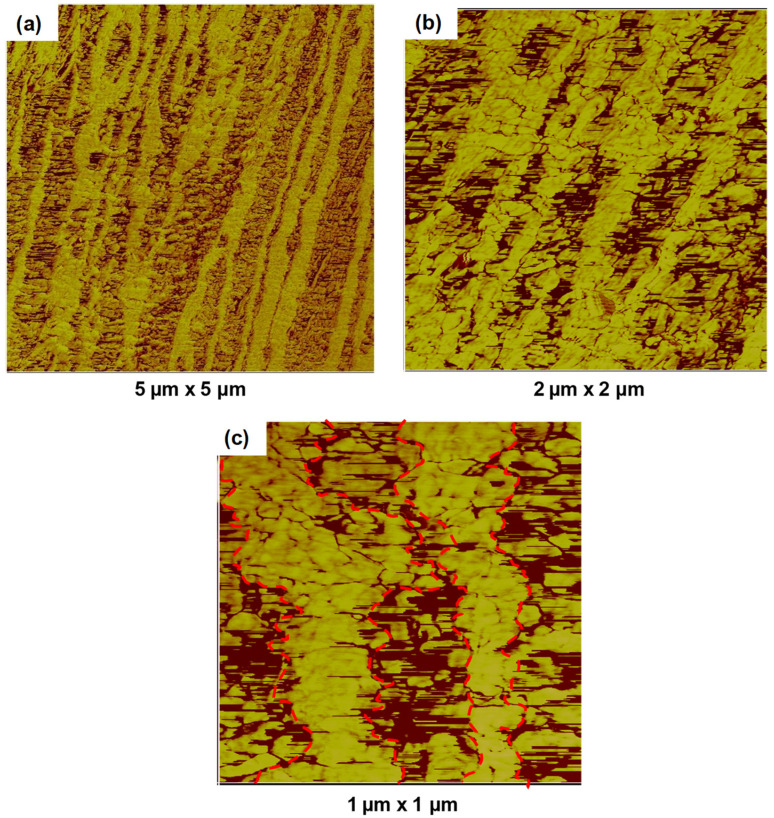
(**a**,**b**) AFM phase images of the domain morphology of the unoriented tapes parallel to the extrusion direction. Domains with lighter colors (higher moduli) are PP; domains with darker colors (lower moduli) are HDPE. Superimposed red dash line in (**c**) highlights the PP-HDPE interfacial region.

**Figure 3 polymers-14-03872-f003:**
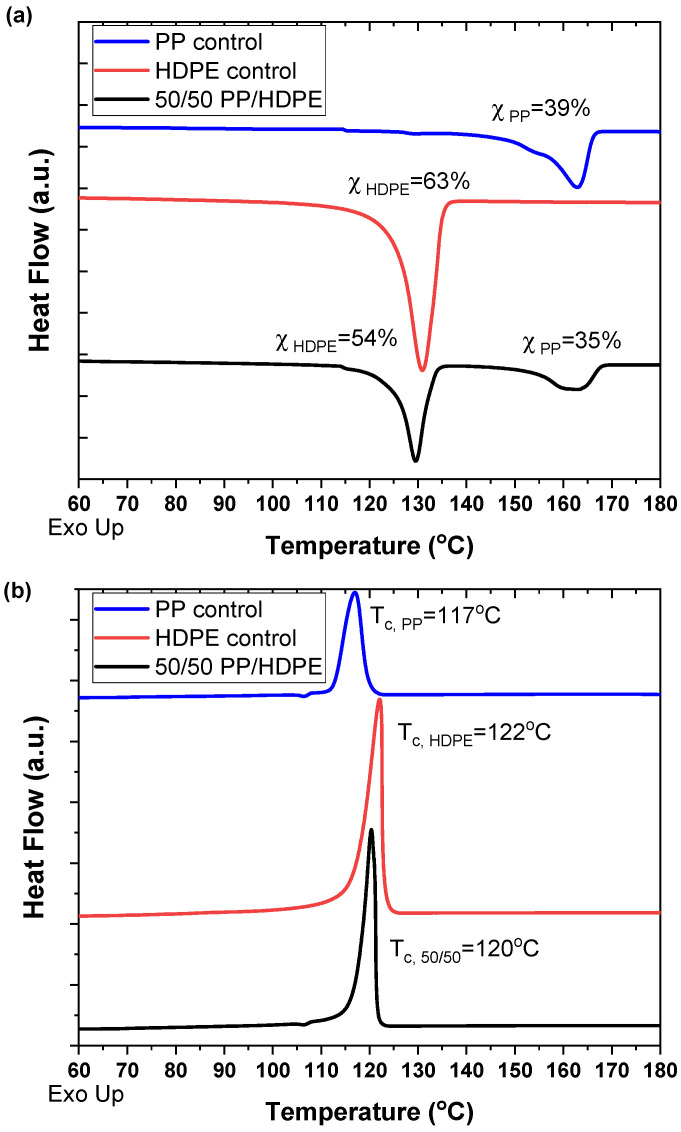
DSC thermographs of unoriented PP control tape, HDPE control tape, and 50/50 PP/HDPE tape; (**a**) heating curve indicating melting peaks; (**b**) cooling curve indicating crystallization peaks.

**Figure 4 polymers-14-03872-f004:**
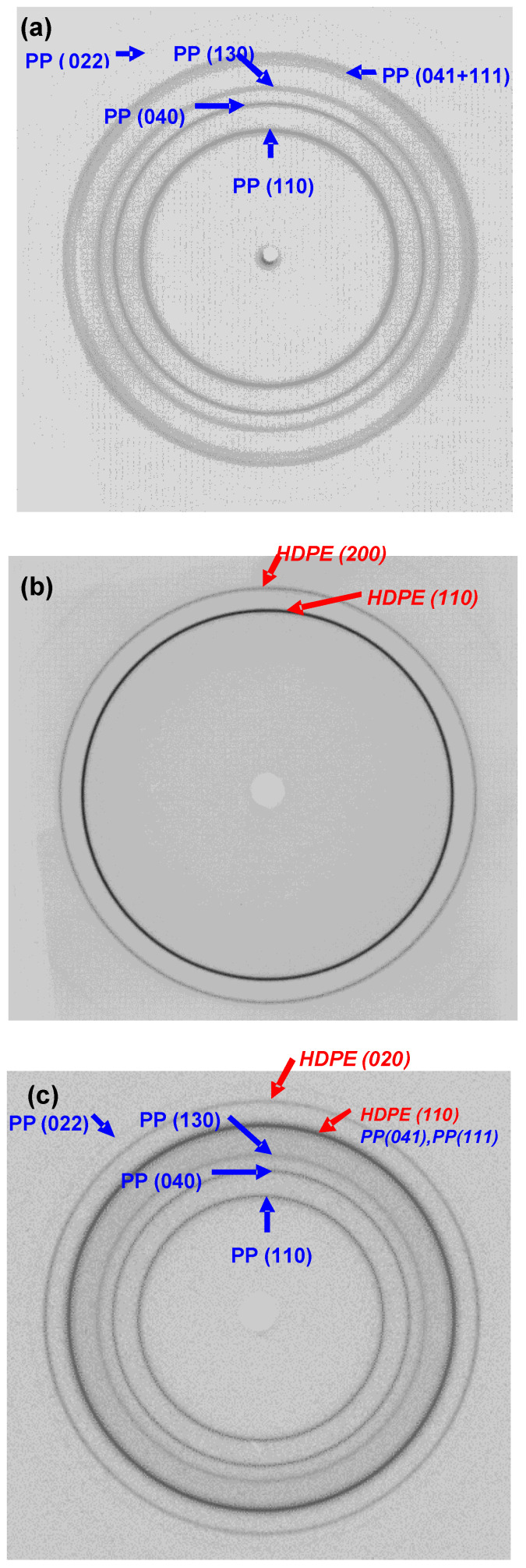
Wide angle X-ray diffraction pattern of unoriented (**a**) PP control tape; (**b**) HDPE control tape; (**c**) 50/50 PP/HDPE tape.

**Figure 5 polymers-14-03872-f005:**
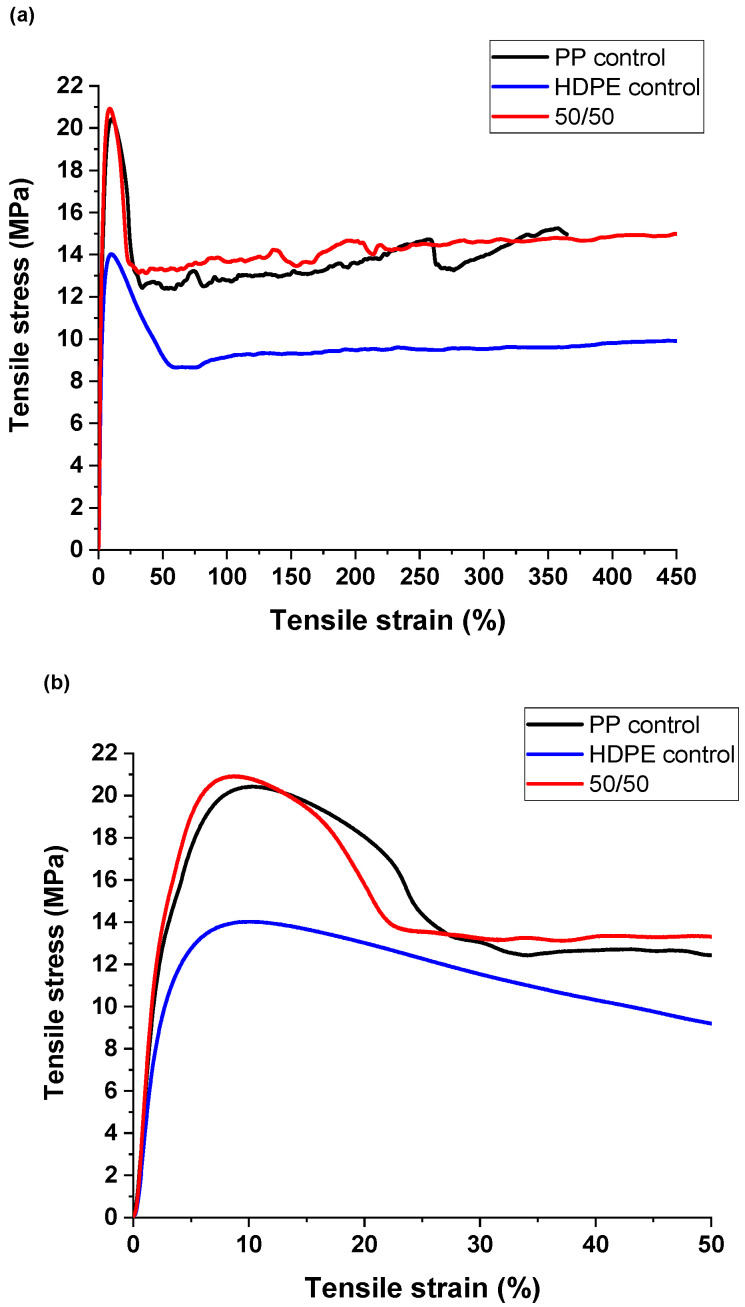
Typical stress–strain curves of unoriented PP control tape, HDPE control tape and 50/50 PP/HDPE tape tested at 100%/min, room temperature; (**a**) overall; (**b**) enhanced 0–5% strain region.

**Figure 6 polymers-14-03872-f006:**
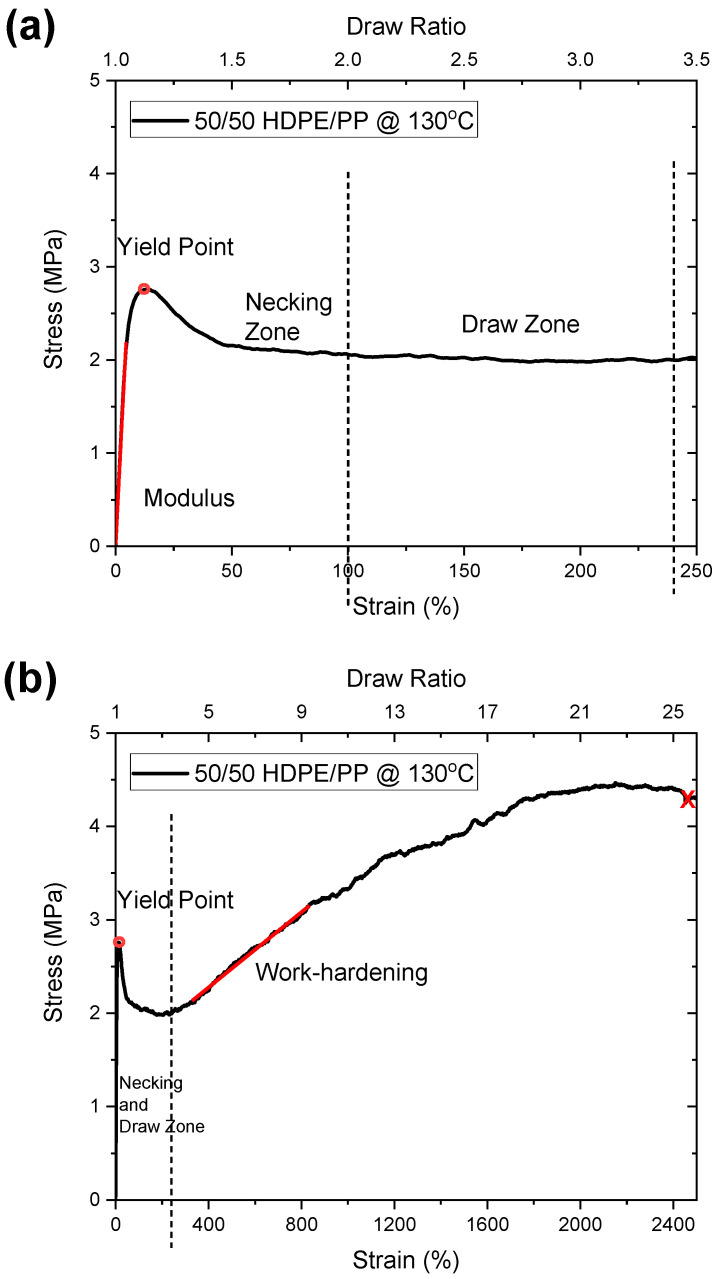
Stress–strain curve during high-temperature orientation of 50/50 PP/HDPE tape at 130 °C; (**a**) is an enhanced portion of (**b**) in the 0–250% region. “X” indicates location of specimen fracture.

**Figure 7 polymers-14-03872-f007:**
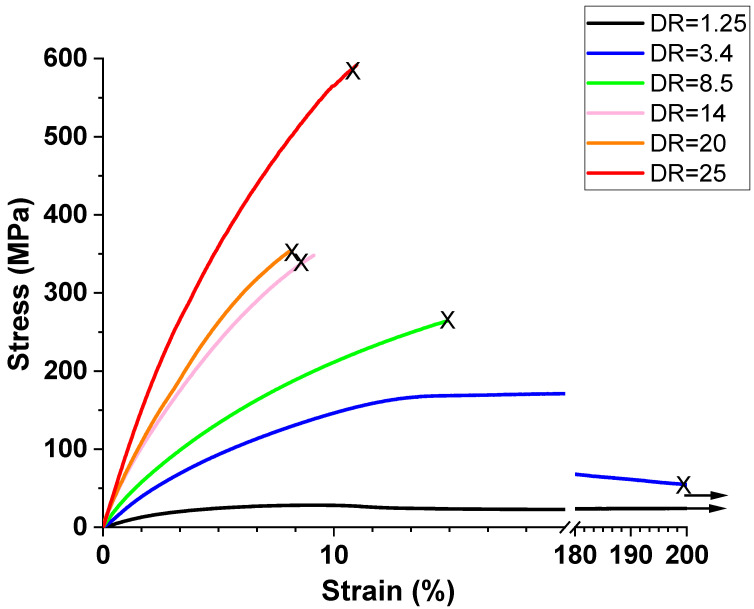
Typical stress–strain curves of oriented 50/50 PP/HDPE tapes at various draw ratios at room temperature. “X” indicates location of specimen fracture.

**Figure 8 polymers-14-03872-f008:**
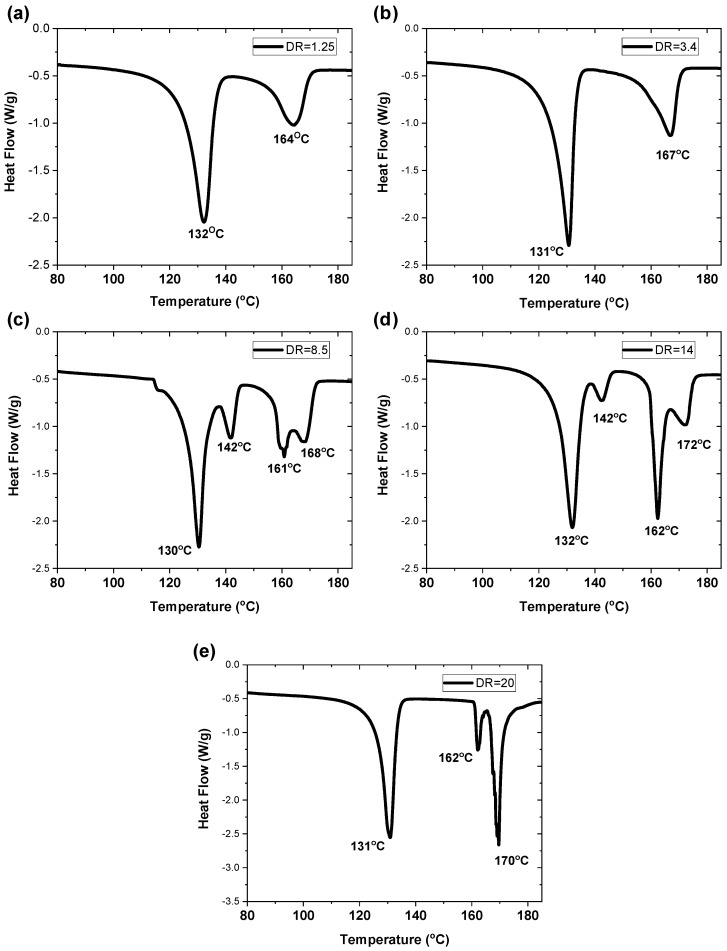
DSC thermographs of oriented 50/50 PP/HDPE tapes at draw ratios of (**a**) 1.25; (**b**) 3.4; (**c**) 8.5; (**d**) 14; and (**e**) 20.

**Figure 9 polymers-14-03872-f009:**
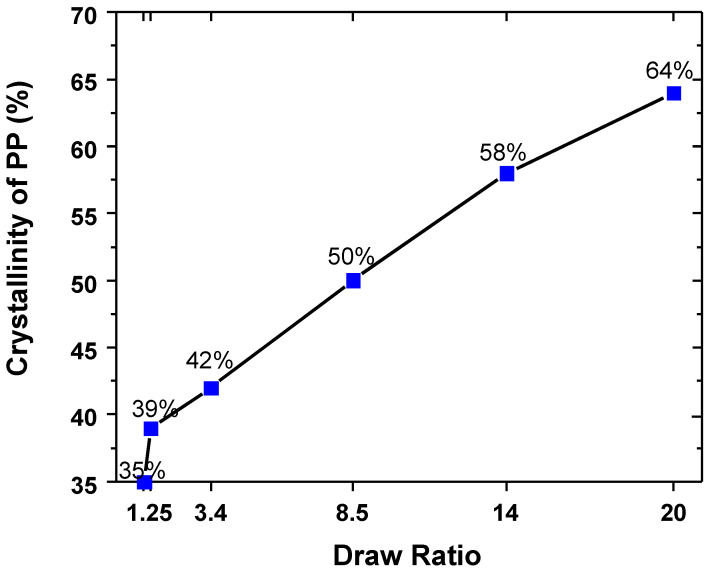
Overall crystallinity of PP as a function of draw ratio.

**Figure 10 polymers-14-03872-f010:**
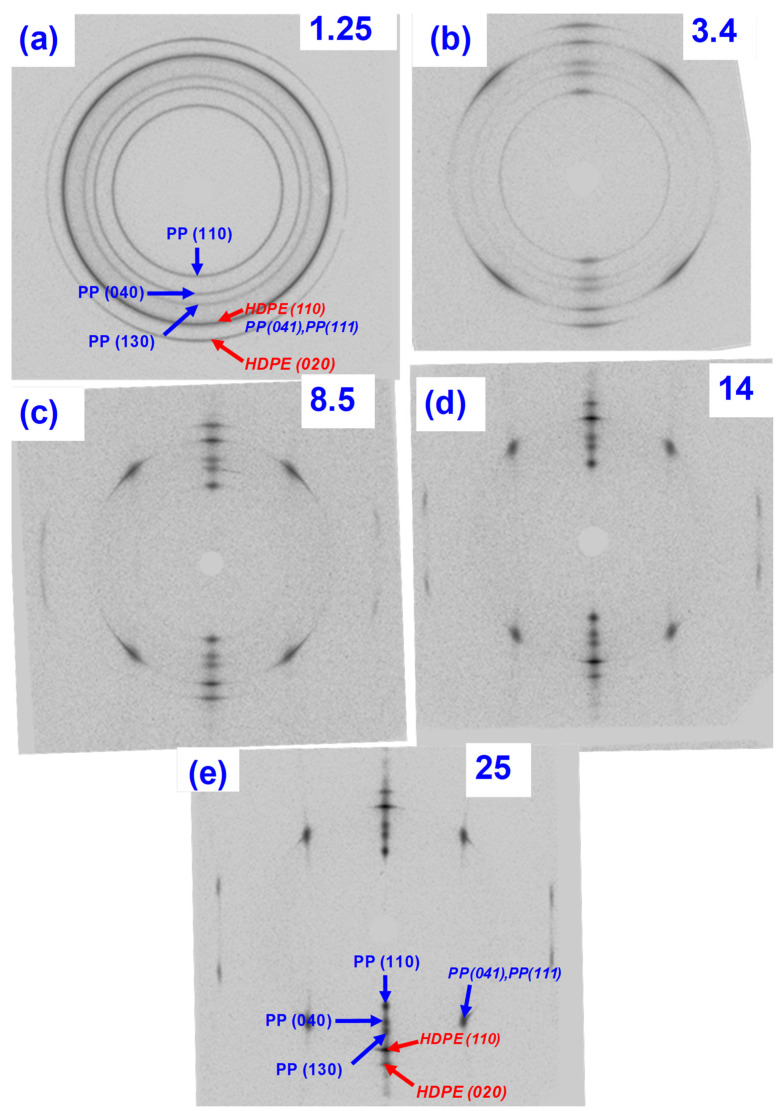
Wide angle X-ray diffraction pattern of oriented 50/50 PP/HDPE. Tapes at draw ratios of (**a**) 1.25; (**b**) 3.4; (**c**) 8.5; (**d**) 14; and (**e**) 25.

**Figure 11 polymers-14-03872-f011:**
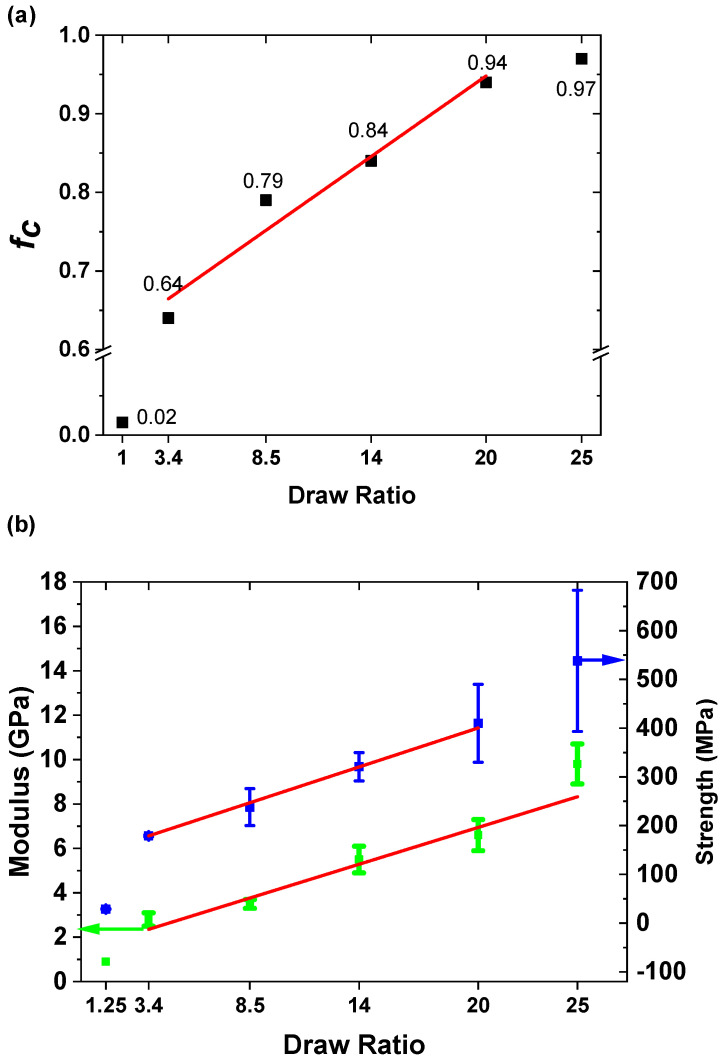
(**a**) Herman’s orientation factor as a function of draw ratio; (**b**) modulus as a function of draw ratio (green) and tensile strength as a function of draw ratio (blue). The mean and standard deviation data are based on the average of five experiments.

**Table 1 polymers-14-03872-t001:** Analysis of stress–strain tests during high-temperature orientation. The mean and standard deviation data are based on the average of three experiments.

	Orientation Temperature (°C)	Modulus (2% Secant) (MPa)	Yield Stress (MPa)	Work-Hardening Slope (MPa)	Fracture Initiation (Draw Ratio)
**50/50 PP/HDPE**	**110 °C**	77 ± 12	7.7 ± 1.2	0.3	20.5 ± 1.0
	**130 °C**	48 ± 5	2.8 ± 0.5	0.2	25.5 ± 0.5
**PP**	**110 °C**	147 ± 14	9.1 ± 1.4	0.3	12 ± 1
	**130 °C**	89 ± 8	4.1 ± 0.3	0.2	29 ± 2
**HDPE**	**110 °C**	41 ± 4	5.9 ± 0.4	0.1	8 ± 1
	**130 °C**	-	-	-	-

**Table 2 polymers-14-03872-t002:** Summary of the mechanical properties of oriented 50/50 PP/HDPE tapes. The mean and standard deviation data are based on the average of five experiments.

Draw Ratio	Modulus (GPa)	Tensile Strength (MPa)	Strain at Break (%)
**1.25**	0.9 ± 0.1	29 ± 1	1400 ± 70
**1.6**	1.0 ± 0.1	41 ± 2	930 ± 100
**3.4**	2.8 ± 0.3	180 ± 2	220 ± 65
**8.5**	3.5 ± 0.2	240 ± 38	13 ± 2
**14**	5.5 ± 0.6	320 ± 29	10 ± 1
**20**	6.6 ± 0.7	410 ± 80	10 ± 2
**25**	9.8 ± 0.9	540 ± 140	10 ± 3

**Table 3 polymers-14-03872-t003:** Comparisons to Other Materials.

Materials	Modulus (GPa)	Tensile Strength (MPa)	Strain at Break (%)
**LDPE/PP Tape (DR = 22)** [8]	6.1 ± 0.3	370 ± 56	---
**PA6/PEO Tape (DR = 6)** [9]	4.2 ± 0.1	240 ± 51	8 ± 1
**Commercial Floss** [9]	0.5 ± 0.3	170 ± 3	42
**PP/HDPE (25/75)** [17]	3.5	60	---
**All PP Composites** [18]	3.8	190	---
**Commercial PP Strapping**	2.2 ± 1.1	300 ± 8	50 ± 17
**50/50 PP/HDPE Tape (DR = 25)** **(This Work)**	9.8 ± 0.9	540 ± 140	10 ± 3

## Data Availability

The data presented in this study are available on request from the corresponding author.

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
