# Peer review of "Oriented Tapes of Incompatible Polymers Using a Novel Multiplication Co-Extrusion Process"

_polymers, 2022, doi:10.3390/polym14183872_

Round 1

Reviewer 1 Report

Dear Authors,

below you can find the review report.

The reviewed article entitled "Oriented Tapes of Incompatible Polymers Using a Novel Multiplication Co-Extrusion Process" concerns the significant problem of using incompatible plastics, such as HDPE and PP for the production of a new material. The subject of creating a blended material of HDPE and PP, presented in the article, is currently widely researched in the world. The research was well prepared, but I found some ambiguities in the article, which I listed in the detailed comments below.

Please check the manuscript editing page carefully: eg. line 158 "" repsectively "should be written:" respectively ".

I suggest that the font size and the thickness of the lines visible in all charts should be standardized.

How were the deviation values for the Modulus, Yield Stress, Fracture Intitiation values shown in Table 1 calculated? The same questions I have are the data seen in table 2 and table 3.

Was a statistical analysis of the obtained results carried out?

Introduction section: Please describe the innovative features of the studies presented in the manuscript. Research into the formation of blends of polyethylene with polypropylene has been conducted for a long time. Information on this subject can be found in many publications (e.g. https://doi.org/10.1002/(SICI)1097-4628(19960606)60:10%3C1527::AID-APP3%3E3.0.CO;2- L.) Research on the properties of structures consisting of microfibers has already been presented in the work https://doi.org/10.1002/pc.20928 of the publication cited by the authors. Please clearly emphasize what new approach to the issue of PP-HDPE tapes creation is proposed by the authors of the manuscript. How does it differ from the method presented in previous publications? What results can be expected from the research.

Figure1 in Figure 1c shows the theoretical tape structure. The theoretical fiber dimensions, see as 7.3x6.2 micrometers, are process based. Please add the information in the manuscript whether and how the actual dimensions of the HDPE/PP fibers were checked?

Author Response

Response to Reviewer #1:

Comment #1:  Please check the manuscript editing page carefully: eg.  line 158 “”repsectively” should be written “respectively”.

Response: Corrected.

Comment #2:  I suggest that the font size and the thickness of the lines visible in all charts should be standardized.

Response: Corrected.

Comment #3:   How were the deviation values for the Modulus, Yield stress, Fracture Initiation values shown in Table 1 calculated?  The same questions I have are the data in table 2 and table 3.

Response:  Multiple specimens were tested for each condition and the mean and standard deviation values were calculated from the results.  An actual test specimen example was used for the respective figures.

Comment #4:  Was a statistical analysis of the obtained results carried out?

Response: Yes. Multiple specimens were tested for each condition and the mean and standard deviation values were calculated from the results.  An actual test specimen example was used for the respective figures.

Comment #5:  Introduction section: Please describe the innovative features of the studies presented in the manuscript.  Research into the formation of blends of polyethylene with polypropylene has been conducted for a long time.  Information on this subject can be found in many publications (e.g., https://doi.org/10.1002/(SICI)1097-46287(19960606)60:10%3C1527::AID-APP3%3E3.0.CO;2-L Research on the properties of structures consisting of microfibers has already been presented in the work https://doi.org/10.1002/pc.20928 of the publication cited by the authors.  Please clearly emphasize what new approach to the issue of PP-HDPE tapes creation is proposed by the authors of the manuscript.  How does it differ from the method presented in previous publications?  What results can be expected from the research.

Response:  The authors could not retrieve:

https://doi.org/10.1002/(SICI)1097-46287(19960606)60:10%3C1527::AID-APP3%3E3.0.CO;2-L

This work differs from others in the literature by the unique multilayer co-extrusion process utilized to produce the multi-domain blend and the recognition of the nano-scale interfacial interactions occurring after orientation of these blends.  These interfacial interactions between two incompatible polymers PP and HDPE arise from the shish-kabob type crystalline morphology induced into both the PP and the HDPE.   This structure-property relationship, in turn, produces enhanced performance of the oriented tape produced by this unique blend morphology.

This has been described in the Abstract, in the Introduction and in the Conclusion.

Specific references have been cited to compare and contrast this unique method of blending two incompatible polymers for enhanced mechanical performance to other process methods attempted.

Comment #6:  Figure 1 in Figure 1c shows the theoretical tape structure.  The theoretical fiber dimensions, see as 7.3 x 6.2 micrometers, are process based.  Please add the information in the manuscript whether and how the actual dimensions of the HDPE/PP fibers were checked?

Response:  As described in this work the multilayer co-extrusion process was used to produce a highly partitioned two-dimensional blend morphology comprising two incompatible polymers PP and HDPE.  The theoretical model only predicts the ideal number of PP and HDPE domains in the tape.  The actual dimensions of the un-oriented domain morphology resulting from this process can be discerned in the AFM image shown in Figure 2.  It is difficult to compare the theoretically predicted domain dimensions to the actual in Figure 2 due to the extremely high number of multiplication and recombination steps undertaken by this process and described in the text of the manuscript. 

Reviewer 2 Report

The research article Oriented Tapes of Incompatible Polymers Using a Novel Multiplication Co-Extrusion Process presents an interesting multiplication co-extrusion process of the fiber-like structure of polyolefin composites.

The article is well-structured, and the results are interesting and well documented. However, some requirements should be performed before publication:

The authors should modify Figure 1. It is of poor quality. Also, the addition of a picture of the co-extrusion system is required. Several schematic representations of the multiplication co-extrusion process is easy to find in literature. Adding pictures of the system is an asset.

The authors should add pictures of the specimen and identify the AFM region.

The authors should include pictures of the tensile specimens (broken or not broken) after the mechanical tests in Figure 5. Please, explain the physical appearance of the tensile specimens (necking, whitening?), is it possible to relate with orientation (chains oriented could be present?) The relationship could complete the analysis performed in Figure 6.

The authors should include pictures of the specimens (post-mortem) in Figure 7. A picture of the mechanical test arrangement could be an asset.

Author Response

Response to Reviewer #2:

Comment #1:  The authors should modify Figure 1.  It is of poor quality.  Also, the addition of a picture of the co-extrusion system is required.  Several schematic representations of the multiplication co-extrusion process is easy to find in literature.  Adding pictures of the systems is an asset.

Response:  Figure 1 has been updated for better quality; the caption has been removed from the figure and placed below the figure. 

Pictures of the multilayer co-extrusion system have already been published in numerous articles.  References to other work from this laboratory [8] to [13] have been cited for those wishing for more information on this process.

Comment #2:  The authors should add pictures of the specimen and identify the AFM region.

Response:  The test specimens were not retained.  There was nothing of interest in the original specimens to identify the nanostructure of interest which is the emphasis of this work.  Only by AFM were we able to identify, in this work, the nano-scale interfacial interactions of the incompatible PP and HDPE domains.

Comment #3:  The authors should include pictures of the tensile specimens (broken or not broken) after the mechanical tests in Figure 5.  please, explain the physical appearance of the tensile specimens (necking, whitening?).  Is it possible to relate with orientation (chains oriented could be present?)  The relationship could complete the analysis performed in Figure 6.

Response:  The data in Figure 5 were based on “control” specimens drawn at “room temperature” and were not taken to fracture.  The critical work was done at elevated temperatures and illustrated as an example in Figure 6.   Multiple specimens were drawn for each material at each temperature and the mean and standard deviation determined.  Figure 6 is a typical stress-strain curve identifying the specific regions for the data in Table 1.  Figure 6 does identify the “necking region”, “draw zone” and the “work hardening region”. 

The samples were not observed to whiten and, in turn, whitening was not mentioned.  

We did not analyze the fracture morphology (i.e., fractography) in this work upon fracture since our focus was on the nano-scale interaction during the drawing stage when drawn at elevated temperatures of the incompatible PP and HDPE domains.  This work as cited in the Abstract, Introduction and Conclusion focused on the interfacial interactions between two incompatible polymers PP and HDPE at the nano-scale.

Reviewer 3 Report

This manuscript reveals the formation of an oriented PP-HDPE blend tape that is stiffer and stronger than the commercially available PP tapes and shows no fibrillation during the drawing. The tape is indicated to have a PP-HDPE co-continuous structure in which both domains are allowed to align in the extrusion direction and form a shish-kebab like morphology after the drawing. The analysis of the crystalline structure of the tape supports the epitaxial crystallization of the polymers having large interfacial area and enhanced adhesion. These data are very interesting and worthy of publication.

This manuscript is well written and can be published after considering the following minor points.

1) The relation of Figures 1 and 2 should be briefly explained in the text for easier understanding of the blend and crystal morphologies.

2) Both Figures 8(c) and 8(d) show a peak at 142 degrees Celsius. How was this peak determined to be due to the extended chain crystals of HDPE? Is there any possibility that this peak is attributed to a different crystal morphism of PP?

3) L. 399-400: “the overall crystallinity of PP increased almost linearly as the draw ratio increased.” Is there any theoretical reason why such a linear relationship is obtained? 

Author Response

Response to Reviewer #3:

Comment #1:   The relation of Figures 1 and 2 should be briefly explained in the text for easier understanding of the blend and crystal morphologies.

Response:  Figure 1 is an “ideal” representation of the multi-layer co-extrusion process producing a micro-scale oriented blend like structure.  This “ideal” figure has been included for better understanding of the multilayer co-extrusion process. 

Figure 2 is the observed morphology obtained from this process and is included to identify the interfacial interactions between the two incompatible polymers, PP and HDPE resulting from this process.

Comment #2:   Both Figures 8(c) and 8(d) show a peak at 142 degrees Celsius.  How was this peak determined to be due to the extended chain crystals of HDPE?  Is there any possibility that this peak is attributed to a different crystal morphism of PP?

Response:  Reference [15] was cited in this section to draw attention to similar observations by Bashir, Odell and Keller.  They observed two melting points for polyethylene, one attributed to an extended chain crystalline structure (~140 degrees Celsius) termed the shish and one attributed to the chain folded crystalline structure (~130 degrees Celsius) termed the kabob.  We are comfortable in drawing this correlation due to our observation of two similar crystalline melting points for polyethylene and the shish-kabob like structure we observed from the AFM analysis.

Comment #3:   L. 399-400: “the overall crystallinity of PP increased almost linearly as the draw ratio increased.”  Is there any theoretical reason why such a linear relationship is obtained?

Response:  We are not aware of any theoretical basis for this apparent linearity and only report it as an observation derived from this work.

Reviewer 4 Report

The authors have done a multiplication co-extrusion process to produce tapes of polypropylene (PP) and high-density polyethylene (HDPE). They optimized the procession conditions to produce oriented tapes with various degrees of softness. The paper is well written and understandable. I have a few minor points, to be addressed before publication:

1-I assume that the increase in the crystallinity of PP with the draw ratio, to be due to chain extension, which allows better packing of chains in the tape. Please comment on this.

2- Because of the anisotropy of the oriented tape, its mechanical properties are expected to be anisotropic. If so, please make it clear in the text and discussions related to Figure 7.

Author Response

Response to Reviewer #4:

Comment #1:   I assume that the increase in the crystallinity of  PP with the draw ratio, to be due to chain extension, which allows better packing of chains in the tape.  Please comment on this.

Response:   We agree.  Our understanding is that the chain extension of the PP contributes to the enhanced crystallinity observed.

Comment #2:   Because of the anisotropy of the oriented tape, its mechanical properties are expected to be anisotropic.  If so, please make it clear in the text and discussions related to Figure 7.

Response:  We agree.  The intent of this work was to investigate the nanoscale inter-crystalline interactions of two incompatible polymers, PP and HDPE.  Due to the induced orientation imparted by the multilayer co-extrusion process and the subsequent uniaxial drawing, the enhanced performance will definitely be in the uniaxial direction.  This work was not focused on investigating the lateral adhesive performance of these materials.

Round 2

Reviewer 1 Report

Dear Authors,

Thank you for your responses. I accept most of them except the lack of statistical analysis and explain what do the whiskers mean on the charts? Does the height of whiskeys present a single standard deviation? 

Author Response

Response to Reviewer #1, Round #2

Comment:  Thank you for your responses. I accept most of them except the lack of statistical analysis and explain what do the whiskers mean on the charts? Does the height of whiskeys present a single standard deviation? 

Response:  The manuscript has been updated to reflect Reviewer #1’s concerns accordingly:

  1. Regarding statistical analysis reference:
  • The caption for Table 1 has been updated with the statement:

“The mean and standard deviation data are based on the average of three experiments.”

  • The caption for Table 2 has been updated with the statement:

“The mean and standard deviation data are based on the average of five experiments.”

  • The caption for Figure 11 has been updated with the statement:

“The mean and standard deviation data are based on the average of five experiments.”

  1. Regarding the “whiskers” noted on the Figures:
  • The caption for Figure 6 has been updated with the statement:

“’X’ indicates location of specimen fracture.”

  • The caption for Figure 7 has been updated with the statement:

“’X’ indicates location of specimen fracture.”
